# Role of the Autism Risk Gene *Shank3* in the Development of Atherosclerosis: Insights from Big Data and Mechanistic Analyses

**DOI:** 10.3390/cells12212546

**Published:** 2023-10-30

**Authors:** Hsiu-Wen Chang, Ming-Jen Hsu, Li-Nien Chien, Nai-Fang Chi, Meng-Chieh Yu, Hsiu-Chen Chen, Yuan-Feng Lin, Chaur-Jong Hu

**Affiliations:** 1Graduate Institute of Clinical Medicine, College of Medicine, Taipei Medical University, Taipei 11031, Taiwan; d118104008@tmu.edu.tw; 2Department of Neurology, Sijhih Cathay General Hospital, New Taipei City 22174, Taiwan; 3Graduate Institute of Medical Sciences, College of Medicine, Taipei Medical University, Taipei 11031, Taiwan; aspirin@tmu.edu.tw; 4Department of Pharmacology, School of Medicine, College of Medicine, Taipei Medical University, Taipei 11031, Taiwansujin99517@gmail.com (H.-C.C.); 5Institute of Health and Welfare Policy, National Yang Ming Chiao Tung University, Taipei 11221, Taiwan; linien.chien@gmail.com; 6Department of Neurology, Neurological Institute, Taipei Veterans General Hospital, Taipei 11267, Taiwan; naifangchi@gmail.com; 7Cell Physiology and Molecular Image Research Center, Wan Fang Hospital, Taipei Medical University, Taipei 11696, Taiwan; 8Department of Neurology, Shuang Ho Hospital, Taipei Medical University, New Taipei City 23561, Taiwan; 9Taipei Neuroscience Institute, Taipei Medical University, Taipei 11031, Taiwan; 10Department of Neurology, School of Medicine, College of Medicine, Taipei Medical University, Taipei 11031, Taiwan

**Keywords:** *Shank3*, autism spectrum disorder, atherosclerosis, lipid metabolism, inflammation, cardiovascular disorder

## Abstract

Increased medical attention is needed as the prevalence of autism spectrum disorder (ASD) rises. Both cardiovascular disorder (CVD) and hyperlipidemia are closely associated with adult ASD. *Shank3* plays a key genetic role in ASD. We hypothesized that Shank3 contributes to CVD development in young adults with ASD. In this study, we investigated whether Shank3 facilitates the development of atherosclerosis. Using Gene Set Enrichment Analysis software (Version No.: GSEA-4.0.3), we analyzed the data obtained from *Shank3* knockout mice (Gene Expression Omnibus database), a human population-based study cohort (from Taiwan’s National Health Insurance Research Database), and a *Shank3* knockdown cellular model. *Shank3* knockout upregulated the expression of genes of cholesterol homeostasis and fatty acid metabolism but downregulated the expression of genes associated with inflammatory responses. Individuals with autism had higher risks of hyperlipidemia (adjusted hazard ratio (aHR): 1.39; *p* < 0.001), major adverse cardiac events (aHR: 2.67; *p* < 0.001), and stroke (aHR: 3.55; *p* < 0.001) than age- and sex-matched individuals without autism did. Shank3 downregulation suppressed tumor necrosis factor-α-induced fatty acid synthase expression; vascular cell adhesion molecule 1 expression; and downstream signaling pathways involving p38, Jun N-terminal kinase, and nuclear factor-κB. Thus, Shank3 may influence the development of early-onset atherosclerosis and CVD in ASD. Furthermore, regulating Shank3 expression may reduce inflammation-related disorders, such as atherosclerosis, by inhibiting tumor necrosis factor-alpha-mediated inflammatory cascades.

## 1. Introduction

Autism spectrum disorder (ASD) is a heterogeneous condition characterized by the impairment of reciprocal social interaction and communication and restricted patterns of interest and activity [1,2]. Research from the Centers for Disease Control and Prevention (CDC) estimated that the prevalence of ASD among adults aged 18 and older in the United States in 2017 was 2.21%, which is similar to the statistics from other recent studies [3,4]. These numbers are in line with the prevalence of ASD in children reported by the CDC’s Autism and Developmental Disabilities Monitoring Network in April 2018, which found that 1 in 56 children has ASD (an approximately three-fold increase from 2000 to 2016; U.S.) [5]. Although health and education services for children with ASD are relatively well established, adequate medical care services are urgently required for adults with ASD. In terms of morbidity and mortality, cardiovascular disorder (CVD) and dyslipidemia are closely associated with autism in adults [6,7,8,9]. Hence, atherosclerosis, which is a key cause of CVD, may be associated with autism.

Atherosclerosis is a chronic inflammatory disease driven by endothelial dysfunction, which promotes the infiltration of immune cells and lipids into the arterial wall, stimulating the production of proinflammatory cytokines and chemokines and subsequently the formation of plaques [10]. This may gradually lead to various medical complications, such as heart attack and stroke. Various factors may contribute to atherosclerosis in individuals with autism, including genetic predisposition, lifestyle (e.g., picky eating habits based on taste and texture preferences) [11,12,13], limited physical activity [14,15,16], and comorbidities. Recent evidence has advanced our understanding of lifestyle–disease interactions. Chronic exposure to environmental stressors such as poor dietary quality, prolonged sitting, and psychosocial stress can affect many pathways associated with atherosclerotic CVD [17,18,19]. A population-based cohort study shows that childhood exposure to cardiovascular risk factors is associated with increased atherosclerosis progression, as assessed through changes in carotid IMT in adulthood over 20 years later [20].

SH3 and multiple ankyrin repeat domains 3 (Shank3) is a multidomain scaffold protein encoded by Shank3 [21,22]. This protein plays a crucial role in synapse formation and dendritic spine maturation [23,24]. Shank3 is a part of the glutamate receptor body (receptosome) that links ionotropic N-methyl-D-aspartate receptors to metabotropic mGlu5 receptors through interactions with the scaffold proteins postsynaptic density-95, guanylate kinase–associated protein, Shank3, and Homer [25]. Haploinsufficiency of the *Shank3* gene causes Phelan–McDermid syndrome, and *Shank3* mutations have been identified with ASD [26]. *Shank 3* is one of the genes associated with ASD and its genetic variants have been associated with clinical neuropsychiatric phenotypes. A previous report demonstrated that two genetic frameshift mutations of *Shank3* resulting in a mis-sense mutation and splicing mutation were identified in 5 of 221 (2.3%) ASD patients and a c. 1304þ48C4T transition (SNP rs76224556), which affects methylated cytosine in the CpG island, was also detected in 17 of 221 (7.7%) ASD patients [27]. It highlights the importance of Shank 3 variants for the effective screening of ASD patients. Moreover, a comparison of the transcriptomes from Shank3-mutant juvenile and adult mice showed opposite patterns and pointed to age, brain region, and gene dosage-differential transcriptomic changes as involved in the altered biological functions and expression of ASD-related genes. As a result, employing Shank3-mutant mice is useful for understanding the mechanisms underlying ASD-related phenotypes in humans [28]. Shank3 mutations are associated with ASD development [29]; however, the underlying molecular mechanisms remain largely unknown. *Shank3* knockout led to social interaction deficits and different phenotypes in mice [26,30]. Although the restoration of *Shank3* expression promoted dendritic spine growth and recovered normal grooming behavior and voluntary social interaction, reduced locomotion, increased anxiety, and social interaction deficits persisted in adult mice with inducible *Shank3* knockout [31,32]. These findings suggest early interventions are effective in restoring behavioral traits in individuals with ASD. Given that multiple investigations have reported a close relationship between CVD, hyperlipidemia, and ASD, we hypothesized that Shank3 associated with ASD can potentially contribute to the initiation and progression of atherosclerosis, the main cause of CVD. Hence, understanding the molecular mechanisms underlying Shank3-related ASD may contribute to understanding the development of atherosclerosis in ASD. Therefore, this study was conducted to identify the signaling pathways that are affected by *Shank3* knockout.

## 2. Materials and Methods

### 2.1. Reagents

Dulbecco’s Modified Eagle Medium, optiMEM, TrypLE, fetal bovine serum, penicillin, streptomycin, and Turbofect in vitro transfection reagent were purchased from Invitrogen (Carlsbad, CA, USA). Antibodies against p65 were purchased from Santa Cruz Biotechnology (Santa Cruz, CA, USA). Antibodies against Shank3 were obtained from Santa Cruz Biotechnology. Antibodies against α-tubulin, phospho-p65 Ser536, Jun N-terminal kinase (JNK), and p38 mitogen-activated protein kinase (MAPK) as well as antimouse and antirabbit immunoglobulin G-conjugated horseradish peroxidase antibodies were purchased from GeneTex Inc. (Irvine, CA, USA). Antibodies against phospho-JNK1/2 and phospho-p38MAPK were purchased from Cell Signaling (Danvers, MA, USA). An enhanced chemiluminescence detection kit was obtained from Millipore (Bellerica, MA, USA). All materials for immunoblotting were purchased from Bio-Rad (Hercules, CA, USA). All other chemicals were obtained from Sigma (St. Louis, MO, USA).

### 2.2. Cell Culture

Mouse brain microvascular endothelial cells (bEnd.3) were purchased from the Bioresource Collection and Research Center (Hsinchu, Taiwan). The cells were maintained in Dulbecco’s Modified Eagle Medium supplemented with 10% fetal bovine serum, 20 mM HEPES, 100 U/mL penicillin G, and 100 μg/mL streptomycin and incubated in a humidified incubator at 37 °C.

### 2.3. Immunoblotting

Immunoblotting was performed as described previously [13]. In brief, bEnd.3 cells were lysed in extraction buffer containing 0.5% NP-40, 10 mM Tris (pH 7.0), 0.05 mM pepstatin A, 0.2 mM leupeptin, 140 mM NaCl, 2 mM phenylmethylsulfonyl fluoride, and 5 mM dithiothreitol. Equal amounts of protein samples were subjected to sodium dodecyl sulfate polyacrylamide gel electrophoresis. The resultant bands were transferred onto a nitrocellulose membrane. After the membrane was blocked with 5% nonfat milk, containing blocking buffer (composition: 5% nonfat milk, 150 mM NaCl, 20 mM Tris-HCl, and 0.02% Tween 20; pH 7.4), for 1 h, it was incubated with specific primary antibodies. After washing, the membrane was incubated with horseradish peroxidase-conjugated secondary antibodies. Immunoreactivity was detected using an enhanced chemiluminescence kit as per the manufacturer’s instructions. Quantitative data were obtained using a computing densitometer and a scientific imaging system (UVP; Biospectrum AC System, UVP, Upland, CA, USA).

### 2.4. Shank3 Knockout

The target gene in bEnd.3 cells was knocked out as described previously [33]. A negative control scramble siRNA and predesigned siRNAs against murine *Shank3* were purchased from Sigma-Aldrich (St Louis, MO, USA). The siRNA oligonucleotides were as follows: *Shank3* siRNA, 5′-CGAUAGAGGAGCAUAGAA-3′; negative control scramble siRNA, 5′-GAUCAUACGUGCGAUCAGA-3′.

### 2.5. Transfection in bEnd.3 Cells

bEND.3 cells were transfected with siRNA using the Turbofect in vitro transfection reagent (Invitrogen, Carlsbad, CA, USA) according to the manufacturer’s instructions.

### 2.6. Reverse-Transcription Quantitative Real-Time Polymerase Chain Reaction

After treatment as indicated, bEND.3 cells were harvested for total RNA isolation. For complementary DNA synthesis, the isolated total RNA was subjected to reverse-transcription quantitative real-time polymerase chain reaction (RT-qPCR). For the DNA synthesis, we used the GoTaq qPCR Master Mix (Promega, Madison, WI, USA) and the StepOne Real-Time PCR system (Applied Biosystems, Grand Island, NY USA). The cycling conditions were as follows: hot-start activation at 95 °C for 120 s, followed by 40 cycles of denaturation at 95 °C for 15 s and annealing/extension at 60 °C for 60 s. The following primers were used to transcribe fatty acid synthase (FASN) and glyceraldehyde-3-phosphate dehydrogenase: mouse FASN—forward, 5′-CACAGTGCTCAAAGGACATGCC-3′; reverse, 5′-CACCAGGTGTAGTGCCTTCCTC-3′; mouse GAPDH—forward, 5′-CCTTCATTGACCTCAACTAC-3′; reverse, 5′-GGAAGGCCATGCCAGTGAGC-3′.

### 2.7. Data Processing and Gene Set Enrichment Analysis

Log2 values corresponding to the fold changes in the mRNA levels of all somatic genes in *Shank3* knockout mice (compared with the mRNA levels in control mice) were derived from the GSE124946 data set using the GEO2R program of the Gene Expression Omnibus database (National Center for Biotechnology Information). A gene list ranked by the log2 fold change was downloaded from the website and subjected to computational simulation (against the hallmark gene sets deposited in the Molecular Signature database) using the Gene Set Enrichment Analysis (GSEA) software (Version No.: GSEA-4.0.3).

### 2.8. National Health Insurance Research Database Analysis

We obtained relevant data (2001–2008) from Taiwan’s National Health Insurance Research Database, which is a population-based health insurance claim database that contains the medical data of almost all residents of Taiwan. Using these data, we conducted a retrospective cohort analysis to identify the likely association between autism and CVD. We first identified the patients who had received a diagnosis of autism when they were aged 30 and 40 years. Each patient with autism was age- and sex-matched with 20 individuals without autism. From the study cohort, we excluded individuals with a history of hyperlipidemia, stroke, or myocardial infraction. The detailed diagnostic codes are presented in Appendix A. The outcomes of interest were hyperlipidemia, stroke, and major adverse cardiovascular events (MACEs, including myocardial infarction, ischemic stroke, and death due to CVD) during the follow-up period. Hyperlipidemia diagnosis was confirmed if the patients had made at least two outpatient and one inpatient diagnostic claims. Myocardial infarction or ischemic stroke was confirmed if the patient had made one inpatient claim. To confirm active diagnosis, patients who had received antiplatelet therapy and undergone computed tomography or magnetic resonance imaging were considered to be diagnosed with myocardial infarction and ischemic stroke, respectively.

### 2.9. Statistical Analysis

Data are presented as the mean ± standard error values of at least five independent experiments. One-way analysis of variance, followed by the Newman–Keuls test when appropriate, was performed to determine the statistical significance of between-group differences.

Survival analysis was performed to identify the association between autism and the outcomes of interest. Kaplan–Meier curves were plotted to estimate the cumulative event rate of the two groups. Cox proportional regression was performed to estimate the hazard ratio and 95% confidence interval (CI). A *p* value of <0.05 was considered to be statistically significant. Statistical analyses were performed using SAS or STAT (version 9.4; SAS Institute, Cary, NC, USA) and STATA or SE 16 (StataCorp, College Station, TX, USA).

## 3. Results

### 3.1. Shank3 Knockout Upregulates the Expression of Genes Associated with Cholesterol Homeostasis and Fatty Acid Metabolism but Downregulates That of Genes Associated with Inflammatory Responses

To elucidate the mechanisms underlying the biological functions of Shank3, we performed computational simulations using GSEA software; the simulations were performed against the Shank3-related gene signature, which is a list of somatic genes in the microarray chip of the GSE124946 data set and ranked by fold changes in the mRNA levels of all tested genes in the brain tissues of *Shank3* knockout mice compared with the levels in control mice. GSEA results indicated that the Shank3-related gene signature was positively correlated with the activity of several hallmark gene sets (Figure 1A) and significantly (*p* < 0.01) correlated with the increased expression levels of genes associated with cholesterol homeostasis (Figure 1B) and fatty acid metabolism (Figure 1C). By contrast, the Shank3-related gene signature was negatively correlated with several hallmark gene sets (Figure 2A) and strongly correlated with the reduced expression of genes associated with inflammatory responses (Figure 2B). A number of studies have revealed that the pro-inflammatory cytokine levels in the blood, brain, and cerebrospinal fluid of autistic subjects differ from those of healthy individuals and different studies have found various results in different tissues [34].

### 3.2. ASD Increases the Risks of Hyperlipidemia, MACE, and Ischemic Stroke

We further analyzed the results of a population-based cohort study conducted among patients with ASD (*n* = 1,416) and matched healthy controls (*n* = 28,320). As shown in Figure 3, autism was associated with elevated risks of hyperlipidemia (aHR: 1.39; 95% CI: 1.24–1.55), MACE (aHR: 2.67; 95% CI: 1.91–3.74), and ischemic stroke (aHR: 3.55; 95% CI: 2.35–5.36). These findings suggest that young adults with ASD are susceptible to early-onset hyperlipidemia, MACE, and stroke.

### 3.3. Shank3 Is Expressed in bEnd3 Cells

Western blotting and RT-qPCR were performed to investigate whether Shank3 is expressed in bEnd.3 cells. As shown in Figure 4, the Shank3 protein (Figure 4A) and mRNA (Figure 4B) were detected in the whole-cell lysates of bEnd.3 cells, but not in murine Raw264.7 macrophages. Thus, we confirmed the expression of Shank3 in bEnd.3 cells.

### 3.4. Shank3 Knockdown Inhibits the Tumor Necrosis Factor-α–Induced Expression of Vascular Cell Adhesion Molecule-1

Because vascular cell adhesion molecule 1 (VCAM-1) plays a major role in atherosclerosis, we investigated whether Shank3 modulates the tumor necrosis factor-α (TNF-α)–induced expression of VCAM-1. Western blotting indicated that TNF-α treatment markedly upregulated the expression of VCAM-1 in bEnd.3 cells; however, this upregulation was absent in *Shank3* knockdown cells (*p* < 0.05) (Figure 4C). RT-qPCR revealed similar findings for VCAM-1 mRNA expression (Figure 4D). Thus, the results indicate that Shank3 may mediate the TNF-α-induced expression of VCAM-1 in bEnd.3 cells.

### 3.5. Shank3 Knockdown Suppresses TNF-α-Induced FASN Expression

*Shank3* knockdown reduces the TNF-α-induced increase in the FASN level in bEnd.3 cells (Figure 5A). RT-qPCR revealed meaningful results for FASN mRNA expression (*p* < 0.05) (Figure 5B). Thus, Shank3 may contribute to TNF-α-induced FASN expression in bEnd.3 cells.

### 3.6. Shank3 Activates TNF-α-Induced Inflammatory Signaling Pathways

We further investigated whether Shank3 modulates the activity of signal transducers, such as p38 MAPK, JNK1/2, and p65 (subunit of nuclear factor [NF]-κB), involved in TNF-α-induced signaling pathways. Western blotting revealed that *Shank3* knockdown did not alter the phosphorylation levels of p38 MAPK, phospho-JNK1/2, or phospho-p65 in bEnd.3 cells compared with the levels in parental cells (Figure 6A). By contrast, TNF-α treatment markedly increased the phosphorylation levels of p38 MAPK (Figure 6B), JNK1/2 (Figure 6C), and p65 (Figure 6C) in bEnd.3 cells; however, these increases were significantly (*p* < 0.05) mitigated after *Shank3* knockdown (Figure 6B–D). These findings suggest that Shank3 modulates various TNF-α-induced inflammatory signaling pathways and thereby influences the development of early-onset atherosclerosis in bEnd.3 cells.

## 4. Discussion

ASD is a common neurodevelopmental disorder prevalent among children. In the present study, the analysis of real-world clinical data obtained from the National Health Insurance Research Database revealed that patients with autism have higher risks of hyperlipidemia, MACE, and stroke than individuals without autism do. The high incidence of early-onset stroke may lead to the development of physical disability, which can be particularly challenging for ASD patients. These risks are increased in patients with ASD, consequently increasing the difficulty in performing daily life functions and elevating emotional stress. Our findings are consistent with those of other studies indicating that CVD and dyslipidemia are closely associated with autism in adults [6,7,9]. The *Shank3* mutation is correlated with ASD; however, the underlying molecular mechanisms remain unknown. Therefore, in this study, we explored the role of Shank3 in atherosclerosis, which is the main cause of CVD. The GSEA results implicated the involvement of Shunk3 in cholesterol homeostasis and fatty acid metabolism. *Shank3* knockdown upregulated the expression of genes associated with cholesterol homeostasis and fatty acid metabolism and downregulated the expression of genes associated with inflammatory responses.

Young people with ASD are at considerably high risks of atherosclerosis-related diseases such as diabetes, obesity, hypertension, and anxiety, all of which are more prevalent in individuals with autism than in healthy individuals. A case–control study evaluating carotid intima–media thickness as an early marker of atherosclerosis in children and adolescents with ASD reported autism symptoms were associated with a high thickness of the carotid intima–media (β = −0.496; *p* = 0.01), indicating the early onset of atherosclerosis in children with ASD [35,36,37,38]. Dyslipidemia, a key risk factor for atherosclerosis [39], can occur at a young age. A study involving 108 adolescent adults with ASD and 206 matched healthy controls reported significantly higher rates of hyperlipidemia in patients with ASD than in healthy controls (31.5% vs. 18.9%) [40]. Thus, patients with autism appear to be highly susceptible to atherosclerosis. Shank3 contributes to cardiac damage because its expression leads to mitochondrial dysfunction and reactive oxygen species production [41]. This finding is consistent with that of the present study in that Shank3 may play an important role in the development of atherosclerosis-associated CVD.

The precise molecular mechanisms underlying atherosclerosis are complex and involve multiple pathways. Accumulating evidence suggests that an increase in the level of the inflammatory cytokine TNF-α, followed by the activation of the NF-κB signaling pathway, plays a pivotal role in the disruption of macro- and microvascular circulation [42]. TNF-α may induce the production of reactive oxygen species and thereby result in endothelial dysfunction, which is typically the first event in atherosclerosis. VCAM-1 is an endothelial adhesion molecule that participates in atherosclerosis by promoting monocyte adhesion, migration, and accumulation in the arterial intima [43,44]. Therefore, VCAM-1 expression may play a major role in the initiation of atherosclerosis. Our findings indicated that VCAM-1 expression was downregulated in TNF-α-treated *Shank3* knockdown cells.

TNF-α receptor 1 is a transmembrane receptor. Upon binding to TNF-α, the receptor activates various intracellular signaling pathways (p38, p65, and JNK), which are involved in inflammatory responses associated with atherosclerosis [42,45]. They are the downstream signaling targets of the TNF-α receptor, and their activation leads to the expression of proinflammatory cytokines, adhesion molecules, and chemokines, which facilitate the recruitment of immune cells into the blood vessel wall. The accumulation of immune cells promotes the growth of atherosclerotic plaques. The activation of these signaling pathways causes oxidative stress, which contributes to endothelial dysfunction and vascular wall damage [42,46,47,48,49]. In the present study, the phosphorylation levels of p38 MAPK, phospho-JNK1/2, and p65 were reduced in TNF-α-treated *Shank3* knockdown cells. Therefore, the results suggest that Shank3 modulates the activities of signaling pathways (p38, JNK1, JNK2, and NF-κB) involved in the development of atherosclerosis-related inflammation.

To the best of our knowledge, this study is the first to report that Shank3-related ASD contributes to atherosclerosis development through the combined effects of cholesterol homeostasis and fatty acid metabolism, and an anti-inflammatory effect, inhibiting the expression of FASN and VCAM-1 and suppressing the activation of the p38, JNK, and NF-κB signaling pathways (Figure 4, Figure 5, Figure 6 and Figure 7). Therefore, the expression of Shank3 in endothelial cells may accelerate the development of atherosclerosis-associated CVD. Our findings indicate that Shank3, a leading ASD candidate gene, modulates the risk of atherosclerosis. Atherosclerosis is the primary pathological basis for CVD; lipid metabolism and inflammation play crucial roles in the development of atherosclerosis. In the case of TNF-α-induced inflammation, *Shank3* knockdown not only significantly inhibited VCAM-1 expression but also partially regulated FASN expression. Notably, *Shank3* knockout significantly influenced lipid metabolism. The lifestyle of patients with ASD may make them susceptible to dyslipidemia, thereby increasing the risk of CVD. Together, the findings suggest that *Shank3* knockdown plays a regulatory role in the development of atherosclerosis in individuals with ASD (Figure 7).

The limitations of this paper include that it cannot directly assess the presence of atherosclerosis in patients with ASD, such as carotid ultrasonography to measure intima–media thickness as a marker of atherosclerosis. And because dyslipidemia is a rather complex condition, this study did not directly conduct cell experiments. Instead, we leveraged NHIRD data to gain insight into real-world conditions. In addition, the experiment utilized mice brain endothelial cells rather than human brain endothelial cells. However, the monoculture model of bEnd3 failed to achieve sufficient tightness as an in vitro BBB permeability model [50].

## 5. Conclusions

Computational simulations performed using GSEA software revealed that *Shank3* knockout potentially activates the signaling pathways involved in cholesterol homeostasis and fatty acid metabolism and suppresses the signaling pathways involved in inflammatory responses. The analysis of real-world clinical data indicated that patients with ASD are at higher risks of hyperlipidemia, MACE, and stroke than healthy individuals are. *Shank3* knockdown substantially suppressed the activation of TNF-α-induced signaling cascades in murine bEND.3 cells. Our findings suggest that young adults with ASD have a relatively high risk of CVD; in other words, they are highly susceptible to atherosclerosis. Shank3 may be involved in cholesterol homeostasis and fatty acid metabolism, potentially leading to atherosclerosis and CVD development. Shank3 also regulates the expression of VCAM-1 and FASN, and Shank3 downregulation suppresses the activation of TNF-α-induced inflammatory signaling pathways. Therefore, the modulation of Shank3 expression may be associated with inflammation-related disorders, such as atherosclerosis, rather than only affecting the brain, through the suppression of TNF-α-mediated inflammation cascades. However, further studies are needed to delineate the roles of inflammatory pathways in Shank3-deficient patients with ASD.

## Figures and Tables

**Figure 1 cells-12-02546-f001:**
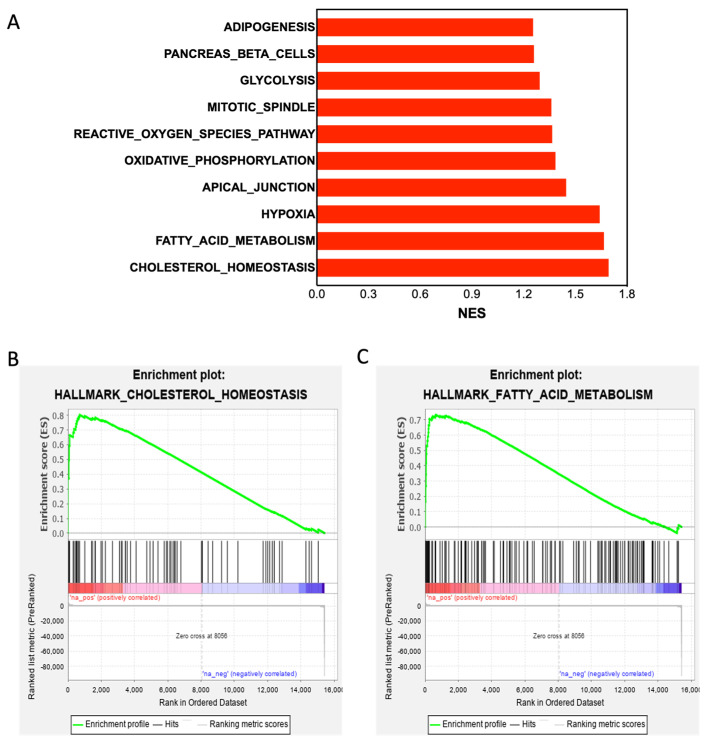
*Shank3* knockout upregulates the expression of genes associated with cholesterol homeostasis and fatty acid metabolism. (**A**) Histogram depicting the positive normalized enrichment scores of simulated Gene Set Enrichment Analysis of the Shank3-related gene signature against the hallmark gene sets. (**B**,**C**) Plots (green line) depicting the enrichment scores derived from the correlation of the Shank3-related gene signature with that of genes associated with cholesterol homeostasis (*p* = 0.001698; (**B**)) and fatty acid metabolism (*p* = 0.001724; (**C**)).

**Figure 2 cells-12-02546-f002:**
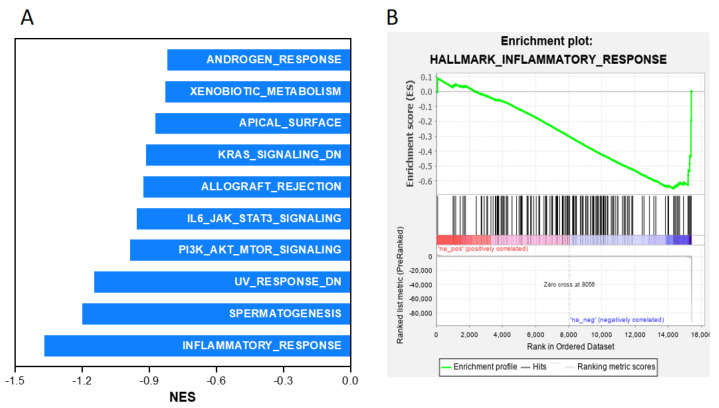
*Shank3* knockout downregulates the expression of genes associated with inflammatory responses. (**A**) Histogram of the negative normalized enrichment scores of simulated Gene Set Enrichment Analysis of the Shank3-related gene signature against the hallmark gene sets. (**B**) Plots (green line) depicting the enrichment score derived from the correlation of the Shank3-related gene signature with that of genes associated with inflammatory responses (*p* = 0.176).

**Figure 3 cells-12-02546-f003:**
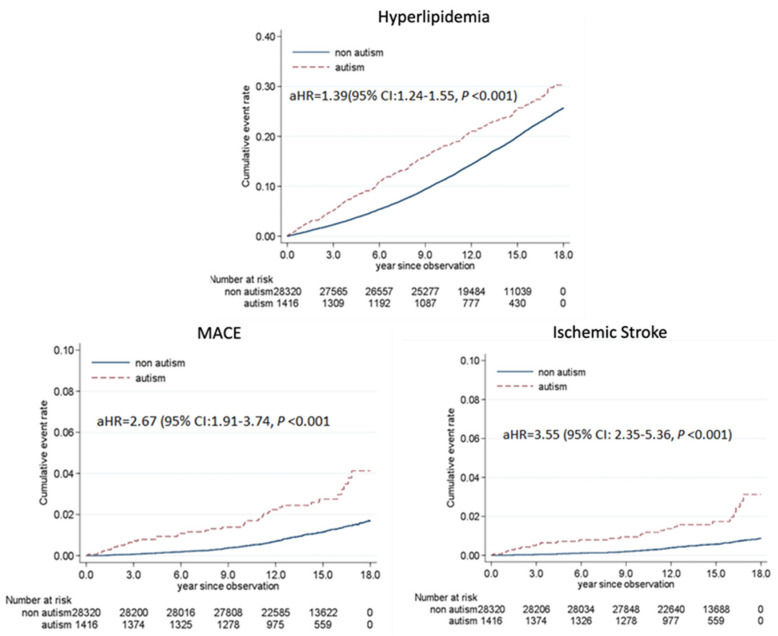
Cumulative curves of hyperlipidemia, major adverse cardiovascular events, and stroke outcomes in patients with autism spectrum disorder and healthy individuals. Autism was associated with a 39% increase in the risk of hyperlipidemia, a 167% increase in that of major adverse cardiovascular events, and a 255% increase in that of stroke; all increases were statistically significant (*p* < 0.001).

**Figure 4 cells-12-02546-f004:**
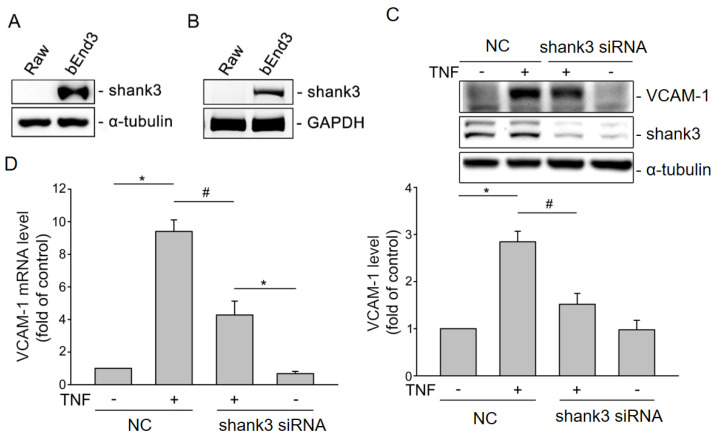
Expression of Shank3 protein and mRNA in murine cerebral endothelial cells (bEnd.3). Western blotting (**A**) and reverse-transcription quantitative real-time polymerase chain reaction (**B**) were performed for measuring the protein and mRNA levels of Shank3 in murine RAW264.7 macrophages and bEnd.3 cells. Shank3 mediated the tumor necrosis factor-α-induced expression of VCAM-1 in bEnd.3 cells. (**C**) Levels of VCAM-1, Shank3, and α-tubulin in the whole-cell lysates derived from nonknockout control and *Shank3* knockout bEnd.3 cells with or without tumor necrosis factor-α simulation. The data presented in the histogram were obtained from three independent experiments. The α-tubulin was used as an internal control. (**D**) Histogram depicting the mRNA levels of measured through reverse-transcription quantitative real-time polymerase chain reaction performed using nonknockout control and *Shank3* knockout bEnd.3 cells with or without tumor necrosis factor-α simulation. The symbols “*” and “#” indicate statistical significance at *p* < 0.05. VCAM-1, vascular cell adhesion molecule.

**Figure 5 cells-12-02546-f005:**
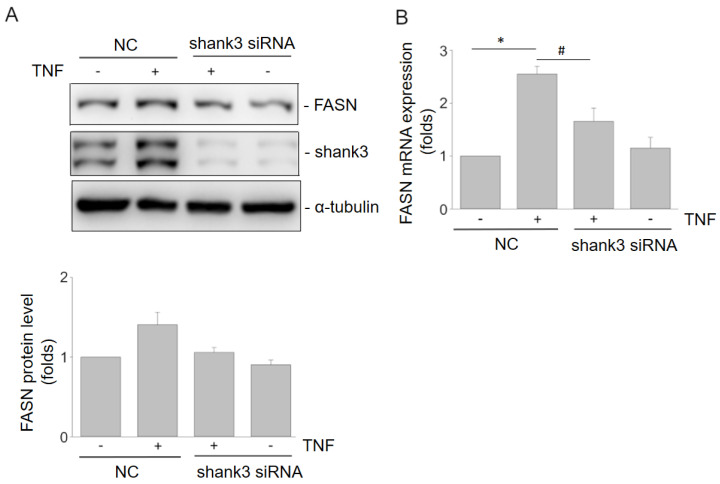
Shank3 mediates tumor necrosis factor-α-induced FASN expression in bEnd.3 cells. (**A**) Levels of FASN, Shank3, and α-tubulin in the whole-cell lysates derived from nonknockout control and *Shank3* knockout bEnd.3 cells with or without tumor necrosis factor-α simulation. The protein levels were measured through Western blotting, and α-tubulin was used as an internal control. The data presented in the histogram were obtained from three independent experiments. (**B**) Histogram depicting the mRNA levels of FASN measured through reverse-transcription quantitative real-time polymerase chain reaction performed using nonknockout control and *Shank3* knockout bEnd.3 cells with or without tumor necrosis factor-α stimulation. The symbols “*” and “#” indicate statistical significance at *p* < 0.05. FASN, fatty acid synthase.

**Figure 6 cells-12-02546-f006:**
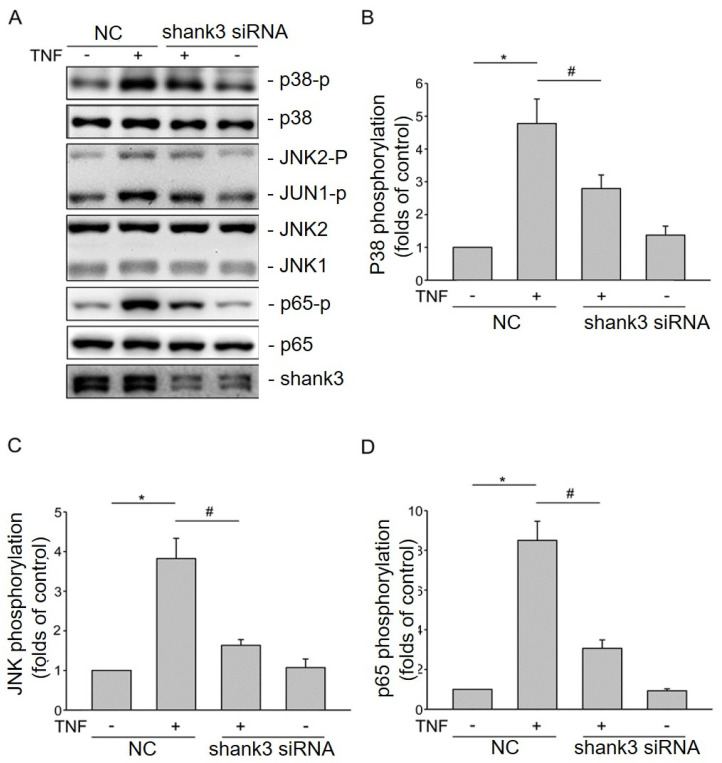
Role of Shank3 in atherosclerosis is mediated through the activation of various signaling pathways (p38, JNK1/2, and nuclear factor-κB). (**A**) Levels of phospho-p38, p38, phospho-JNK1/2, JNK1/2, phospho-p65, p65, and Shank3 in the whole-cell lysates derived from nonknockout control and *Shank3* knockout bEnd.3 cells with or without tumor necrosis factor-α simulation. The protein levels were measured through Western blotting. (**B**–**D**) Histogram depicting the levels of phospho-p38 (**B**), phospho-JNK1/2 (**C**), and phosphor-p65 normalized by the corresponding expression levels of total protein in three independent experiments. The symbols “*” and “#” indicate statistical significance at *p* < 0.05. JNK, Jun N-terminal kinase.

**Figure 7 cells-12-02546-f007:**
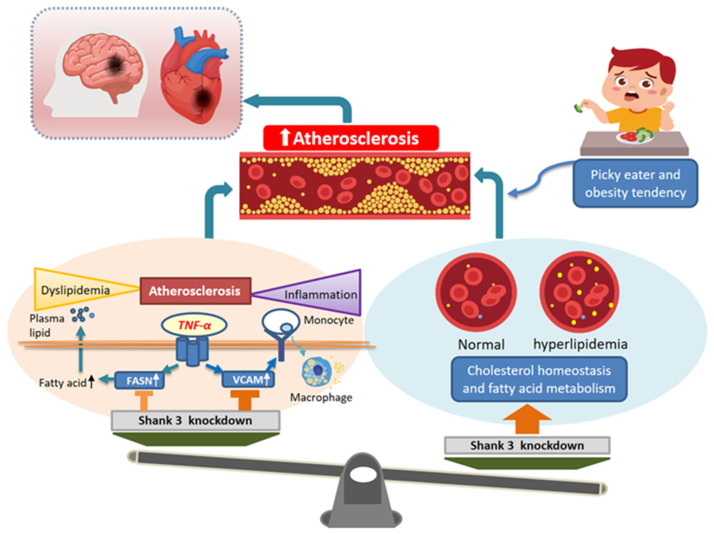
Results of the comprehensive analysis of the role of *Shank3* in the development of atherosclerosis in individuals with autism spectrum disorder. Lipid metabolism and inflammation are two important factors leading to vascular sclerosis. Shank3 plays a role in mediating the tumor necrosis factor-α-induced inflammatory response. In addition to inhibiting the expression of vascular cell adhesion molecule 1, Shank3 regulates the expression of fatty acid synthase. However, because of the significant effect of *Shank3* knockdown on lipid metabolism and various lifestyle factors, individuals with ASD are susceptible to atherosclerosis, which eventually leads to cardiovascular disease.

## Data Availability

The data presented in this study are available in the article.

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
