# Peer review of "Role of the Autism Risk Gene Shank3 in the Development of Atherosclerosis: Insights from Big Data and Mechanistic Analyses"

_cells, 2023, doi:10.3390/cells12212546_

Round 1

Reviewer 1 Report

Comments and Suggestions for Authors

The authors aimed to prove the relationship between Shank3 and atherosclerosis. They used animal cells (mouse) to build their model. Such a design raises some questions, that need answering. 

I would like to start with the Introduction. Authors suggest that the prevalence of ASD has increased in not only children (an 54 approximately 3-fold increase; United States) [3] but also adults (a 13-fold increase; Statis- 55 tics, Ministry of Health and Welfare 2008-2020, Taiwan). The meaning of this sentece is unclear. It shows that ASD could be acquired during lifetime which is not true. Maybe the authors relate to increasing number of ASD diagnosis due to better diagnosis tools/programs? This should be cleared out. 

Similarly, it is risky to link atherosclerosis directly to ASD through one gene, because non-neurotypical individuals can have lower level of physical activity or food selectivity. These aspect should be described within the Introduction with adequate references. 

It should be emphasized that authors used animal cellular model. 

Limitation of the study should also mention the differences between animal and human cellular model. 

Concusions should be re-written taking into account this study findings, being careful with interpretation. 

Author Response

Response to Reviewer 1 Comments

Point 1. Authors suggest that the prevalence of ASD has increased in not only children (an 54 approximately 3-fold increase; United States) [3] but also adults (a 13-fold increase; Statis- 55 tics, Ministry of Health and Welfare 2008-2020, Taiwan). The meaning of this sentece is unclear. It shows that ASD could be acquired during lifetime which is not true. Maybe the authors relate to increasing number of ASD diagnosis due to better diagnosis tools/programs? This should be cleared out.

Response: Thank you for your critical comments and constructive suggestions. An analysis of the number of adults with autism spectrum disorder is based on applications for disability certification. There may be two main reasons for the increase:

  1. Increased prevalence due to increased awareness of the disease. The use of health insurance in Taiwan may also be related to diagnostic criteria.
  2. Disability certification was re-established in 2012 based on the International Classification of Functioning, Disability and Health (ICF) developed by the World Health Organization (WHO). The increase in applications may be attributed to changes in application criteria.

Therefore, we looked to the literature to understand the prevalence of ASD in adults. We have included the description as follows.

“Research from the Centers for Disease Control and Prevention (CDC) estimated the prevalence of ASD among adults aged 18 and older in the United States in 2017 was 2.21%. This study addresses the lack of data on ASD adults. These numbers are consistent with the prevalence of ASD in children reported by the CDC's Autism and Developmental Disabilities Monitoring Network in April 2018, which found that 1 in 56 children has ASD (an approximately 3-fold increase from 2000 to 2016; U.S.) [3].” (page 2,line: 53-58)

Reference list

  1. Autism Research 12: 370–374, 2019; Autism prevalence and outcomes in older adults. doi: 10.1002/aur.2080.
  2. J Autism Dev Disord. 2020 Dec; 50(12): 4258–4266. National and State Estimates of Adults with Autism Spectrum Disorder

doi: 10.1007/s10803-020-04494-4

Point 2. Similarly, it is risky to link atherosclerosis directly to ASD through one gene, because non-neurotypical individuals can have lower level of physical activity or food selectivity. These aspect should be described within the Introduction with adequate references.

Response: We appreciate this valid critique and constructive suggestions. Shank3 is one of the well-known risk factors for autism. Cardiovascular disease and dyslipidemia are closely associated with autism. We found additional references explaining this nontraditional risk factor for atherosclerosis. The new paragraph looks like this. 

“Recent evidence has advanced our understanding of lifestyle–disease interactions. Chronic exposure to environmental stressors such as poor dietary quality, prolonged sitting, and psychosocial stress can affect many pathways associated with atheroscle-rotic CVD [1-3]. A population-based cohort study shows that childhood exposure to cardiovascular risk factors is associated with increased atherosclerosis progression, as assessed by changes in carotid IMT in adulthood over 20 years later [4].” (page 2, line 71-76)

Reference list

  1. Katharina Lechner , Clemens von Schacky, Amy L McKenzie et.al. Lifestyle factors and high-risk atherosclerosis: Pathways and mechanisms beyond traditional risk factors. Eur J Prev Cardiol. 2020 Mar;27(4):394-406. doi:10.1177/2047487319869400.
  2. Taotao Wei , Junnan Liu , Demei Zhang et. The Relationship Between Nutrition and Atherosclerosis Taotao Wei , Junnan Liu , Demei Zhang. Front Bioeng Biotechnol. . 2021 Apr 19:9:635504. doi: 10.3389/fbioe.2021.635504.
  3. Stefan Acosta, Anna Johansson and Isabel Drake. Diet and Lifestyle Factors and Risk of Atherosclerotic Cardiovascular Dis-ease—A Prospective Cohort Study. Nutrients 2021, 13, 3822.
  4. Markus Juonala, Jorma S.A. Viikari, Mika Ka¨ho¨nen et. Life-time risk factors and progression of carotid atherosclerosis in young adults: the Cardiovascular Risk in Young Finns. European Heart Journal (2010) 31, 1745–1751.study. doi:10.1093/eurheartj/ehq141

Point 3. It should be emphasized that authors used animal cellular model. Limitation of the study should also mention the differences between animal and human cellular model.

Response: Thank you for your critical comments and constructive suggestions. As suggested, we have added additional information about the limitations of the study. The new paragraphs are shown as follows.

“In addition, the experiment utilized mice brain endothelial cells rather than human brain endothelial cells. However, the monoculture model of bEnd3 failed to achieve sufficient tightness as an in vitro BBB permeability model.” (page12, line406-409)

Reference list

  1. Yang S, Mei S, Jin H, Zhu B, Tian Y, Huo J, et al. (2017) Identification of two immortalized cell lines, ECV304 and bEnd3, for in vitro permeability studies of blood-brain barrier. PLoS ONE 12(10): e0187017.

Reviewer 2 Report

Comments and Suggestions for Authors

The manuscript by Chang et al. describes the analysis of an autistic risk factor, Shank3's roles in Atherosclerosis, using a cohort study, which is one of the interesting studies. 

  1. The manuscript keywords are precise.
  2. It is an interesting study using a significant cohort and its control. 
  3. The title matches with their study.

The authors need to improve the following sections.

Introduction

  1. Introduction line 56, the prevalence of ASD in adults requires more precise reference.
  2. Line 60, Add the reference for the Atherosclerosis and ASD study. Some reviews are available; if the authors could not find it, add this statement as their hypothesis.
  3. Shank3 is one of the well-known autistic risk factors. Authors should add more details about the variant types associated with ASD and describe human phenotypes with animal model studies.
  4. Line 190 mentions a few hallmark genes for each cholesterol, fatty acid metabolism, and inflammation.
  5. Mention the P in a proper style in the entire manuscript.
  6. In the figures, put the same font size.
  7. Figure 4, panels A and B look like Western, in the legend mentioned as Western and qPCR. Please correct it. 
  8. 4C, treatment of TNF in NC and Shank3 siRNA expression levels seem similar, but the quantification shows that significantly reduced. The authors should indicate the transparent panel.
  9. All westerns show a precise depletion level of shank3 except the 6A panel.

Author Response

Response to Reviewer 2 Comments

Point 1 Introduction line 56, the prevalence of ASD in adults requires more precise reference.

Response: Thank you for your critical comments and constructive suggestions. An analysis of the number of adults with autism spectrum disorder is based on applications for disability certification. There may be two main reasons for the increase:

  1. Increased prevalence due to increased awareness of the disease. The use of health insurance in Taiwan may also be related to diagnostic criteria.
  2. Disability certification was re-established in 2012 based on the International Classification of Functioning, Disability and Health (ICF) developed by the World Health Organization (WHO). The increase in applications may be attributed to changes in application criteria.

Therefore, we looked to the literature to understand the prevalence of ASD in adults. The new paragraphs are shown as follows.

“Research from the Centers for Disease Control and Prevention (CDC) estimated the prevalence of ASD among adults aged 18 and older in the United States in 2017 was 2.21%, which is similar to the statistics from other recent studies. These num-bers are in line with the prevalence of ASD in children reported by the CDC's Autism and Developmental Disabilities Monitoring Network in April 2018, which found that 1 in 56 children has ASD (an approximately 3-fold increase from 2000-2016; U.S.).

(page 2,line: 53-58)

Reference list

  1. Autism Research 12: 370–374, 2019; Autism prevalence and outcomes in older adults. doi: 10.1002/aur.2080.
  2. J Autism Dev Disord. 2020 Dec; 50(12): 4258–4266. National and State Estimates of Adults with Autism Spectrum Disorder

doi: 10.1007/s10803-020-04494-4

Point 2 Line 60, Add the reference for the Atherosclerosis and ASD study. Some reviews are available; if the authors could not find it, add this statement as their hypothesis.

Response: Thank you for your suggestions and comments. Since no reference to this was found, this hypothesis has been described in the original text, as explained below.

“Given that multiple investigations have reported a close relationship between CVD, hyperlipidemia, and ASD, we hypothesized that Shank3 associated with ASD can po-tentially contribute to the initiation and progression of atherosclerosis, the main cause of CVD.” (page 3, line 103-105)

Point 3 Shank3 is one of the well-known autistic risk factors. Authors should add more details about the variant types associated with ASD and describe human phenotypes with animal model studies.

Response: Thank you for this critical comments and constructive suggestions. As suggested, we have added the more detailed information regarding the association between Shank 3 variants and ASD and the description of human phenotypes with animal model studies in Introduction section. The new paragraphs are shown as follows.

“Haploinsufficiency of the SHANK3 gene causes Phelan-McDermid syndrome, and SHANK3 mutations have been identified with ASD. Shank 3 is one of the genes associated with ASD and its genetic variants have been associated with clinical-neuropsychiatric phenotypes. Previous report demonstrated that two genetic frameshift mutations of Shank 3 resulting in a missense mutation and splicing mutation were identified in 5 of 221 (2.3%) ASD patients and a c. 1304þ48C4T transition (SNP rs76224556) which affects a methylated cytosine in a CpG island was also detected in 17 of 221 (7.7%) ASD patients (1). It highlights an important of Shank 3 variants for the effective screening of ASD patients. Moreover, the comparison of the transcriptomes from the Shank 3-mutant juvenile and adult mice showed an opposite patterns and pointed to age, brain region, and gene dosage-differential transcriptomic changes which involve in the altered biological functions and expression of ASD-related genes (2). As a result, employing the Shank3-mutant mice is useful for understanding the mechanisms underlying the ASD-related phenotypes in human.” (page 2. Line 82-95)

Reference List

1 Shigeo Uchino 1, Chikako Waga. SHANK3 as an autism spectrum disorder-associated gene Brain Dev. . 2013 Feb;35(2):106-10.

  1. Boccuto L, Lauri M, Sarasua SM et al. Prevalence of SHANK3 variants in patients with different subtypes of autism spectrum disorders. Eur.J.Hum.Genet. 2013;21:310-6.
  2. Yoo T, Yoo YE, Kang H, Kim E. Age, brain region, and gene dosage-differential transcriptomic changes in Shank3-mutant mice. Front Mol.Neurosci. 2022;15:1017512.

4 Line 190 mentions a few hallmark genes for each cholesterol, fatty acid metabolism, and inflammation.

Response: Yes. Through GSEA analysis, it was found that shank 3 knockout is related to several marker genes of fatty acid metabolism and inflammation. These genes were significantly correlated with the expression of cholesterol- and fatty acid-related genes and were inversely correlated with inflammation.

5 Mention the P in a proper style in the entire manuscript.

Response: Appreciate for your comments and suggestion. Appropriate style corrections have been made to p throughout the manuscript.

Point 6 In the figures, put the same font size.

Response: Thanks for your valid comments and suggestion. The words in the figure have been modified to the same font size.

7 Figure 4, panels A and B look like Western, in the legend mentioned as Western and qPCR. Please correct it.

Response: Thanks for this valid comment. Fig. 4A represents Western data, while Fig. 4B represents data obtained through RT-PCR, focusing on mRNA levels. The presented images are traditional agarose gel data that has been scanned.

Point 8 4C, treatment of TNF in NC and Shank3 siRNA expression levels seem similar, but the quantification shows that significantly reduced. The authors should indicate the transparent panel.

Response: We appreciate this valid critique. To quantify western blot bands, the densitometry is performed using ImagePro software. Based on the quantification results derived from 7 independent experiments (shown below), shank3 siRNA significantly reduced TNF-a-induced VCAM-1 expression.  The VCAM-1 level in the NC group differs from that in the shank3 siRNA group, as shown in Figure 4C.  

folds

NC

NC+TNF

shank3 siRNA

Shank3 siRNA+TNF

1.0

3.0

1.9

1.1

1.0

2.3

0.4

0.4

1.0

3.1

1.3

0.3

1.0

2.5

1.2

1.1

1.0

2.4

1.7

1.9

1.0

2.6

2.0

0.9

1.0

4.0

2.2

1.2

9 All westerns show a precise depletion level of shank3 except the 6A panel.

Response: Thanks for this comment. The 6 A panel has been corrected to show a precise depletion level of shank3.

Round 2

Reviewer 1 Report

Comments and Suggestions for Authors

Thank you for addressing all comments.